# Impact of E484Q and L452R Mutations on Structure and Binding Behavior of SARS-CoV-2 B.1.617.1 Using Deep Learning AlphaFold2, Molecular Docking and Dynamics Simulation

**DOI:** 10.3390/ijms241411564

**Published:** 2023-07-17

**Authors:** Yanqi Jiao, Yichen Xing, Yao Sun

**Affiliations:** School of Science, Harbin Institute of Technology (Shenzhen), Shenzhen 518055, China; abigcat1204@gmail.com (Y.J.); 22s058083@stu.hit.edu.cn (Y.X.)

**Keywords:** molecular dynamics simulation, molecular docking, AlphaFold2, double mutant variant

## Abstract

During the outbreak of COVID-19, many SARS-CoV-2 variants presented key amino acid mutations that influenced their binding abilities with angiotensin-converting enzyme 2 (hACE2) and neutralizing antibodies. For the B.1.617 lineage, there had been fears that two key mutations, i.e., L452R and E484Q, would have additive effects on the evasion of neutralizing antibodies. In this paper, we systematically investigated the impact of the L452R and E484Q mutations on the structure and binding behavior of B.1.617.1 using deep learning AlphaFold2, molecular docking and dynamics simulation. We firstly predicted and verified the structure of the S protein containing L452R and E484Q mutations via the AlphaFold2-calculated pLDDT value and compared it with the experimental structure. Next, a molecular simulation was performed to reveal the structural and interaction stabilities of the S protein of the double mutant variant with hACE2. We found that the double mutations, L452R and E484Q, could lead to a decrease in hydrogen bonds and higher interaction energy between the S protein and hACE2, demonstrating the lower structural stability and the worse binding affinity in the long dynamic evolutional process, even though the molecular docking showed the lower binding energy score of the S1 RBD of the double mutant variant with hACE2 than that of the wild type (WT) with hACE2. In addition, docking to three approved neutralizing monoclonal antibodies (mAbs) showed a reduced binding affinity of the double mutant variant, suggesting a lower neutralization ability of the mAbs against the double mutant variant. Our study helps lay the foundation for further SARS-CoV-2 studies and provides bioinformatics and computational insights into how the double mutations lead to immune evasion, which could offer guidance for subsequent biomedical studies.

## 1. Introduction

The severe acute respiratory syndrome coronavirus 2 (SARS-CoV-2) has posed major health, political, economic, and social problems with unprecedented consequences [1]. Today, it is still spreading among people, with its spike (S) protein continuously mutating [2]. The S protein is usually the primary target of neutralizing antibodies since it is exposed to the virus surface and mediated into host cells [3]. It can be cleaved into two non-covalently associated subunits designated S1 and S2 during the biosynthesis in the infected cells [4,5]. The S1 subunit contains the receptor-binding domain (RBD) and N-terminal domain (NTD) which binds to the human angiotensin-converting enzyme 2 (hACE2) and recognizes attachment factors [6]. The S2 subunit consists of a fusion peptide and other machinery and undergoes large-scale conformational changes to drive the viral and host membrane fusion. Antibodies that bind to specific sites on the RBD, NTD, or fusion machinery can interfere with receptor attachment or membrane fusion [5]. Up to now, many biochemical studies have been devoted to the development of various vaccines against SARS-CoV-2 [7,8]. However, the recent variants still show tendencies toward higher transmissibility, virulence, frequency of reinfection, and resistance to the monoclonal and polyclonal antibodies [9,10].

The B.1.617 lineage, including the B.1.617.1 (Kappa), B.1.617.2 (Delta), and B.1.617.3 variants, were first identified in India in late 2020 to early 2021 [11]. The B.1.617.1 variant S contains T95I, G142D, E154K, L452R, E484Q, D614G, P681R, and Q1071H replacements [12]. The B.1.617.2 variant S has T19R, G142D, E156G, L452R, T478K, D614G, P681R, and D950N replacements as well as the absence of residues 157 and 158 [12]. The B.1.617.3 variant S contains T19R, L452R, E484Q, D614G, P681R, and D950N [13]. The B.1.617.1 and B.1.617.2 variants were designated as a Variant of Interest (VOI) and a Variant of Concern (VOC), respectively, while B.1.617.3 was not classified as either a VOI or VOC, since few reports of this variant have been submitted to date [14]. VOCs or VOIs are those SARS-CoV-2 mutants that are more virulent and infectious than their wild types (WTs) (Wuhan-Hu-1). 

The mutations localized at RBD (residues 319 to 541) and NTD (residues 14 to 305) are the primary targets of neutralizing monoclonal antibodies (mAbs) in vaccinating individuals, hence the increasing concerns about the efficacies of available vaccines and therapeutic mAbs against these variants [15]. For example, research on the B.1.617 lineage showed that E484K/Q and L452R had significant effects on increasing the fusion activity. When attempting to neutralize the Kappa S protein with Pfizer vaccine serum, E484K was found to reduce neutralization by a factor of 10, and E484Q had a slightly more modest but significant effect. However, combining E484Q with L452R caused a statistically significant loss of sensitivity [16]. In another study, the Covaxin vaccine (BBV152) was observed to cause a two-fold reduction in neutralizing efficacy against the Delta variant [17]. In general, previous studies on the B.1.617 lineage showed a slight decrease in the neutralizing activity of the vaccines compared to the WTs [18]. It was also reported that L452R and E484Q might play a role in the increase in the propagation rate, but the molecular mechanisms behind these actions are still unclear [19]. 

In the past, many studies focused on exploring the structures of S proteins of different SARS-CoV-2 variants through experimental methods, which have greatly advanced our understanding of the influences of mutations on altering the structures and functions of SARS-CoV-2 [9,20]. However, uncovering the structures of S proteins for a large number of SARS-CoV-2 variants remains a challenge due to the high experimental cost and rapid increase in variation [21]. In addition, it is quite difficult to model the SARS-CoV-2 S structure containing artificially designated mutations. Fortunately, AlphaFold2 (DeepMind, Google) emerged as a computational method that can rapidly predict protein structures based on gene sequences with quite high accuracy [22,23,24]. Since the structure of the S protein of the double mutant variant with E484Q and L452R mutations is not directly available from the RCSB Protein Data Bank (PDB) [25], we adopted AlphaFold2 to model the S protein with these double mutations. The high-precision structure was verified by comparison with the experimental structure and by using the pLDDT (predicted local distance difference test) of the AlphaFold2 built-in algorithm [26]. We further investigated the dynamics evolutions of structures and energy variations of S proteins of WT and the double mutant variant with hACE2. In addition, molecular docking analyses were conducted on S1 RBDs with hACE2 as well as three prevalent mAbs.

## 2. Results

### 2.1. Structure Predictions of S Proteins for the WT and Double Mutant Variant

To evaluate the predictive structure of the S protein of the double mutant variant, we analyzed the value of the pLDDT calculated using AlphaFold2, which is a value representing the accuracy of the corresponding prediction [23,27]. A pLDDT value above 70 indicates that the structure obtained can be considered as a reliable predictive structure [26]. Therefore, we selected Rank 0 (pLDDT: 78.46) and Rank 1 (pLDDT: 78.98) from all the predicted structures of the double mutant variant (Figure 1a). The pLDDT (Ranks 0–4) values of the full-length S protein, S1 RBD, and S1 NTD of the double mutant variant can be found in Appendix A. The mean pLDDT (Ranks 0–4) values were higher than 70.01, 79.02, and 86.61, respectively. We also compared the two predicted structures of the S protein of the WT with the experimental S protein (PDB ID: 7DF4) to evaluate the validity of our prediction. The template modeling (TM), maximum Subset (MaxSub), and Global Distance Test (GDT)-TS scores were calculated and compared between AlphaFold2-predicted and experimental S proteins. Normally, TM, MaxSub, and (GDT)-TS scores are used to compare two protein structures based on a given residue equivalence [28], evaluate the quality of protein structure prediction [29], and find the protein similarities in protein structures [30], respectively. The obtained TM score >0.81, MaxSub scores >0.62, and (GDT)-TS scores >0.60 (Appendix A) demonstrated the high confidence of our predicted structures. The structure comparison showed that the S protein of the predicted WT was highly similar to 7DF4, with the low root-mean-square deviation (RMSD) values of 1.886 Å and 1.453 Å for Rank 1 and Rank 0, respectively (Figure 1a). The structure comparison also showed that the S1 RBD of the predicted WT was highly similar to experimental 6M0J, with the low RMSD values of 1.37 Å and 1.485 Å for Rank 1 and Rank 0, respectively (Figure 1c).

The S1 RBD is usually divided into three parts, including the N1 (residues 333–438), receptor-binding motif (RBM, residues 438–506), and N3 (residues 507–539) [31]. The RMSD values of N1, RBM, and N3 between the double mutant variant and WT were 1.2 Å, 0.7 Å, and 5.5 Å, respectively (Figure 1d). The N3 domain was significantly different in the double mutant variant and the WT, while the N1 and RBM domains showed high similarities. In addition, we observed a significant increase in the positive electrostatic surface potential in the double mutant variant (Figure 1b). In terms of the S protein, the double mutant variant was observed to have higher positive electrostatic potential than the WT, which may promote RBD to interact with the negatively charged hACE2. We further compared the electrostatic properties of the S1 structures of 7DF4, the predicted double mutant variant Rank 0 and Rank 1. The double mutations, L452R (aliphatic L452 to positively charged R452) and E484Q (polar uncharged E484 to positively charged Q484), occurring in the S1 RBD could change the electrostatic properties of the S protein significantly (Figure 1b). In order to further investigate the effects of structural changes on binding antibodies, the RMSD values of five loops of S1 NTDs of the double mutant variant and WT were calculated. The five loops included the N1 loop (residues 14–26), N2 loop (residues 67–79), N3 loop (residues 141–156), N4 loop (residues 177–186), and N5 loop (residues 246–260). The results showed that the RMSD values of N2 and N5 were both larger than 4.0 Å, while the RMSD values of N1, N3, and N4 were 2.530, 3.131, and 1.937, respectively (Figure 1e).

### 2.2. Results of RMSD and RMSF Values, Hydrogen Bonds and Interaction Energy via MD Simulations

We found papers regarding the S protein of SARS-CoV-2 with MD simulations for 150–200 ns [32,33,34,35]; therefore, the MD run was conducted for 200 ns in our study. The mean RMSD values with standard deviations of the double mutant variant-hACE2 and WT-hACE2 complexes were 0.51 ± 0.06 Å and 0.50 ± 0.05 Å, respectively, as shown in Figure 2A. We also analyzed the root-mean-square fluctuation (RMSF) values of the amino acid residues in S1 RBD of the B chains in the WT and double mutant S proteins (Figure 2B). This showed that the RMSF values of residues in the S1 protein of the double mutant spike were generally larger than those of the WT. In particular, the largest RMSF value of residues in S1 RBD of the double mutant spike was 0.94 nm, located at residue 477, which was significantly larger than the largest value of 0.53 nm located at residue 479 in S1 RBD of the WT. The interaction energy between the S protein of the double mutant variant and hACE2 reached a stable energy state of −968 ± 288 KJ/mol during the 150–200 ns, which was higher than the corresponding value of −1292 ± 239 KJ/mol between the WT and hACE2 (Figure 2D). During the first 150 ns, the number of hydrogen bonds between the S protein of the WT and hACE2 showed a larger fluctuation than its counterpart. During the last 150–200 ns, the number of hydrogen bonds in the S protein of the double mutant variant and hACE2 reached 11 ± 7, which was lower than that of the WT and hACE2 (18 ± 7) (Figure 2C).

### 2.3. The Molecular Architectures of S Proteins of WT and Double Mutant Variant after 200 ns MD Simulations

Subsequently, we analyzed the structure of the S1 RBD of the WT after the 200 ns MD simulation and found that it rotated counterclockwise by 12°, and the overall length of the S1 RBD decreased from 108 Å to 99 Å (Figure 3a). Contrarily to the results of the WT spike, the S1 RBD of the double mutant variant rotated clockwise by 16°, and the overall length increased from the initial 82 Å to 94 Å after the 200 ns MD simulation (Figure 3c). By comparing the structural changes in the S1 RBDs for the two S proteins, we noticed that the S1 RBD of the double mutant variant expanded by stretching downward, while the S1 RBD of the WT contracted upwards during the binding processes with hACE2. In addition, we investigated the structural changes in S1 NTDs for the double mutant variant and WT after the 200 ns simulations. From the side view of the S1 NTDs (Figure 3a,c), we found that the S1 NTDs of the WT and double mutant variant showed no obvious expansion or contraction over the simulation time. From the top view of the S1 NTD of the WT (Figure 3b), it can be seen that the NTDs in the B chain, C chain and D chain rotated 16° clockwise, 8° clockwise, and 1° clockwise, respectively. In terms of the double mutant variant, the NTDs in the B chain, C chain, and D chain rotated 29° clockwise, 7° clockwise, and 9° counterclockwise, respectively (Figure 3d). In order to carefully examine the expansion or contraction of the S1 NTD after the simulation, the distance between the Ser256 of NTDs in different spike chains (Ser256.B-Ser256.C: BC; Ser256.C-Ser256.D: CD; Ser256.D-Ser256.B: DB) were measured. Ser256 is on the surface of the S protein and is located close to the edge of the NTD. For the double mutant variant, the surface area of the triangle reduced from 11,259 Å^2^ to 9515 Å^2^, while the surface area of the triangle increased from 12,778 Å^2^ to 13,775 Å^2^ for the WT.

### 2.4. The Binding of Double Mutant Variant to hACE2 on Host Cells

The crystal structure of the SARS-CoV-2 S1 RBD and hACE2 complex (PDB ID: 6M0J) was used as the template. The S1 RBDs of the double mutant variant (Rank 1) and WT were docked with hACE2 in two protein–protein docking tools, i.e., HADDOCK 2.4 [36] (https://wenmr.science.uu.nl/haddock2.4/, accessed on 16 January 2023) and HDOCK [37] (http://hdock.phys.hust.edu.cn/, accessed on 16 January 2023). The results showed that the docking scores of the S1 RBD of the double mutant variant with hACE2 (HADDOCK score: −123.7 ± 4.0, HDOCK score: −347.29) were lower than those of the WT with hACE2 on both platforms (HADDOCK score: −116.1 ± 1.6, HDOCK score: −310.19) (Table 1), indicating the better affinity ability of the S1 RBD of the double mutant variant to hACE2. The interaction interface can be found in Figure 4. The study of 2.5 Å interacting residues in UCSF Chimera [38] (https://www.rbvi.ucsf.edu/chimera, accessed on 16 January 2023) between the double mutant variant-hACE2 complex suggests that the two mutated residues (L452R and E484Q) show interactions with hACE2 and elevate their binding affinity (Appendix A). According to the structural analysis, seven residues (Asn487, Glu484, Gln493, Tyr453, Tyr449, Gly496, and Thr500) in the S1 RBD of WT and hACE2 were in interaction. In terms of the double mutant variant, eight residues (Thr500, Gly502, Tyr505, Gly496, Tyr449, SER494, Lys417, and Ala475) in S1 RBD were in interaction with hACE2 (Appendix A). Docking simulations were performed to quantify the interactions between the hACE2 and S1 RBDs. The results show that amino acid mutations affect the docking scores between the hACE2 and S1 RBD of the double mutant variant, causing the changes in the interaction (Table 1 and Figure 4). In addition, the double mutant variant-hACE2 complex showed the lower RMSD from the overall lowest-energy, Van der Waals energy, electrostatic energy, Z-score, and ligand RMSD values than those of the WT-hACE2 complex, suggesting the stronger binding affinity and higher structural stability of the double mutant variant-hACE2 complex (Table 1). The larger buried surface area of the double mutant variant with hACE2 (1834.8 ± 27.4 Å^2^) indicated its larger accessible surface area than its counterpart WT with hACE2. Most notably, a significant difference in electrostatic energy could be observed between the WT-hACE2 and the double mutant variant-hACE2 complexes. The electrostatic energy of the WT-hACE2 was −221.5 ± 11.0, while the value was −248.1 ± 29.2 for the double mutant variant (Table 1), indicating that the mutations of positively charged R452 and Q484 could reduce the electrostatic energy. In general, these results reveal that the double amino acid mutations in the S1 RBD could significantly alter the structure of the S1 RBD and its interaction with hACE2. 

### 2.5. The Binding of Double Mutant Variant to Neutralizing mAbs

To infer the binding affinity and neutralizing ability of the double mutant variant with the currently known mAbs, docking of the Fab regions (heavy and light chains) of three mAbs (casirivimab, bamlanivimab, and etesevimab) was completed on the S1 RBDs of the WT and double mutant variant (Table 2). For the WT–Bamlanvimab, the HDOCK score (−125.996) and HADDOCK score (−275.49) were both lower than those of the double mutant variant–Bamlanvimab (HDOCK score: −107.493, HADDOCK score: −181.94). For the WT–Casirivimab, the HDOCK scores (−121.918) and HADDOCK scores (−407.74) were also lower than those of the double mutant variant–Casirivimab (HDOCK score: −94.374, HADDOCK score: −94.374). In terms of the double mutant variant–Etesevimab, the HDOCK score (−108.941) and HADDOCK score (−184.8) were both higher than the corresponding values of WT–Etesevimab (HDOCK score: −123.423, HADDOCK score: −263.07). These results demonstrate that all the three mAbs show lower binding affinity to S1 RBD due to the double mutations, indicating that vaccines appear to be less effective against the double mutant variant. Although the docking characteristics of the three mAbs are quite different (Figure 5), Etesevimab was found to be more effective in preventing the double mutant variant than the other two mAbs (Table 2). Thus, the data suggest that double mutations in the S1 RBD provide an advantage to the virus by decreasing the binding affinity with mAb and reducing the neutralization of these antibodies.

## 3. Discussion

In this study, we successfully predicted the structure of the S protein of the SARS-CoV-2 mutant variant containing two important mutations, E484Q and L452R. The reliability of the predicted S protein was verified by using the AlphaFold2-calculated [26] pLDDT value and by comparing it with the experimental structure of the S protein [39]. To the best of our knowledge, there are quite limited studies on exploring the changes in specific residues in the S1 RBD structure of the SARS-CoV-2 Kappa variant [40,41]. This helps lay the foundation for further SARS-CoV-2 studies and complement experimental structural biology. Our results also provide bioinformatics and computational insights into how the double mutations lead to immune evasion, which could offer guidance for subsequent biomedical studies [42,43]. Furthermore, the methods presented can be used to conveniently monitor mutations and structural variations for more SARS-CoV-2 variants [9,44]. 

Compared with the WT, the S protein of the double mutant variant showed less hydrogen bonds and higher binding energy with hACE2, indicating their lower interaction stability. In addition, the structural stability of the S protein of the double mutant variant was found to be worse than its counterpart due to its much larger RMSF values. However, the results of molecular docking tell another story. The docking scores of the S1 RBD of the double mutant variant with hACE2 were lower than those of WT with hACE2 on both the HADDOCK and HDOCK platforms, indicating the better affinity ability between the S1 RBD of the double mutant variant and hACE2. The study of 2.5 Å interacting residues between the double mutant variant-hACE2 complex suggested that the two mutated residues (L452R and E484Q) showed interactions with hACE2 and elevated their binding affinity. Our docking results are consistent with the work of Kumar et al., which suggests that the mutations at L452R, T478K, and E484Q could increase the stability and intra-chain interactions in the S protein based on the MM/GBSA binding free energy calculations [45]. However, in our study, we propose that the double mutations of L452R and E484Q lead to the decrease in hydrogen bond interactions and higher interaction energy between the S protein and hACE2, demonstrating the lower structural stability and the worse binding affinity of the double mutant variant with hACE2 in the long dynamic evolutional process, even though the molecular docking shows the lower binding energy score of the S1 RBD of double mutant variant-hACE2. Based on analyses of molecular architectures after the 200 ns MD simulations, we propose a plausible mechanism that the effect of double mutations on infectibility is due to the corresponding conformational transition of the S protein. As we have shown, the S1 RBD of the double mutant variant extends inward and upward, making the S protein of the double mutant variant more likely to interact with hACE2. The changes in E484Q and L452R may affect the opening of RBD. Also, the NTD exhibits flexibility, and its conformational dynamics is mainly reflected between the closed and open states. When the NTD moves downward or outward, it facilitates the opening of RBD and releases the strength of the protopolymer interaction between S1 and S2. By analyzing the MD results, the NTD of the S protein of the double mutant variant extended outward more than that of the WT, causing the S1 domain to move away from S2 and the double mutant spike to be more infectious. Lastly, molecular docking to three approved mAbs showed the reduced binding affinity of the double mutant variant with mAbs, suggesting a reduced neutralization ability of the mAbs against the double mutant variant. It might be interpreted that the decrease in the binding energy score of the double mutant variant can lead to stabilization and interference with neutralizing antibody interactions [46].

## 4. Materials and Methods

### 4.1. Structure Modeling with AlphaFold2

The structure prediction of the S protein was based on the AlphaFold2 model published in *Nature* [23]. The model ran with all genetic databases and eight ensembles. The AlphaFold2 pipeline of reasoning and source could be obtained under the open-source license https://github.com/deepmind/alphafold (accessed on 23 November 2022). Our preset parameters were the same as those used in CASP14 [23], and the predict_structure function was adopted for protein prediction to obtain the 5 best predicted protein models. The mirror databases used in our study included BFD, Mgnify Cluster, UniRef90, Uniclust30, Protein Database (PDB), and PDB70 [47]. The sequence of the S protein of the double mutant variant was downloaded from GISAID (http://gisaid.org, accessed on 15 November 2022).

### 4.2. Calculation of the Similarity between Experimental and Predicted Structures

The experimental structure of S protein was downloaded from PDB (PDB ID: 7DF4) (https://www.rcsb.org/, accessed on 15 November 2022) [48]. The 3D protein structures were visualized using PyMol [49]. Similarity difference tests (pLDDT) between the experimental and predicted structures were evaluated using template modeling (TM) scores, maximum subset (MaxSub) scores, global distance test GDT-TS scores, root mean square deviation (RMSD), and predicted local distances [30,50]. TM scores [28], MaxSub scores [29], and GDT-TS scores were based on TM-align (https://zhanggroup.org/, accessed on 31 December 2022) [51] for online services calculated between the experimental structure and the predicted structure of RMSD and pLDDT, respectively, via PyMol and AlphaFold2 software calculations [52].

### 4.3. Molecular Dynamics Simulations

The Cryo-EM structure of the S protein of the WT with hACE2 (PDB ID: 7DF4) was obtained from PDB. The WT-hACE2 complex was prepared by removing water molecules and eutectic ligands, leaving only amino acid residues. The L452R and E484Q double mutant variant-hACE2 complex was established by mutating residue Leu452 to Arg452 and Glu484 to Gln484 of the WT ear using UCSF Chimera 1.15 [38]. The AMBER99SB-ILDN force field [53] was used to parameterize proteins, implemented in the GROMACS 5.1.2 software package [54]. The particle grid Ewald (PME) method was used to calculate the remote electrostatic interactions. Each simulation was performed in an explicit water solvent via a TIP3P water box [55]. The protein was placed in the center of the cube box, 1.2 nm from the box boundary. Thirty-three Na+ ions were introduced into the water tank to neutralize the charge of the entire system. Energy minimization and equilibrium were divided into the following three steps: (i) We used the steepest descent algorithm to minimize the entire system containing ions, solvents, and proteins to 100,000 steps. (ii) We balanced the S proteins and hACE2 to 310 K (normal human temperature, 100 ps, V-rescale) with backbone constrained. (iii) We integrated NPT at constant pressure (1 bar) and temperature (310 K) for 500 ps. Finally, a 200 ns MD run was performed with all constraints removed. The results were analyzed using Gromacs built-in tools and our internal scripts.

### 4.4. Molecular Docking Studies of hACE2 with S1 RBDs

The crystal structure of the S1 RBD-hACE2 complex (PDB ID: 6M0J) was obtained from PDB. The S1 RBD of the double mutant variant was the Rank 1 predicted via AlphaFold2. The S1 RBD of the WT was obtained by deleting hACE2 from 6M0J. The hACE2 was obtained in a similar way by deleting the S1 RBD from 6M0J and adding polar hydrogen atoms in the Discovery Studio Visualizer. HDOCK (https://life.bsc.es/pid/pydockweb, accessed on 16 January 2023) and HADDOCK 2.4 (https://wenmr.science.uu.nl/haddock2.4/, accessed on 16 January 2023) were used to conduct the molecular docking. We studied 2.5 Å interacting residues in UCSF Chimera [38] (https://www.rbvi.ucsf.edu/chimera, accessed on 16 January 2023) to infer complex interactions between important residues.

### 4.5. Molecular Docking Studies of mAbs with S1 RBDs

To infer the effects of currently available mAbs on the S1 RBDs of the WT and double mutant variant, the Fab regions of three mAbs (casirivimab, bamlanivimab, and etesevimab) were docked to the S1 RBDs. The S1 RBD of the double mutant variant was the Rank 1 structure predicted using AlphaFold2. The three antibodies casirivimab (PDB ID: 6XDG), bamlanivimab (PDB ID: 7KMG), and etesevimab (PDB ID: 7C01) were docked to S1 RBDs using HDOCK and HADDOCK 2.4 servers.

## 5. Conclusions

In this study, we used AlphaFold2 to obtain the structure of the S protein of the double mutant variant containing two important mutations, E484Q and L452R. According to the results of the MD simulation and molecular docking, the two mutations led to a decrease in hydrogen bond interactions and higher interaction energy between the S protein and hACE2, demonstrating the lower structural stability and the worse binding affinity of the double mutant variant with hACE2 in the dynamic evolutional process, even though the molecular docking shows the lower binding energy score of the S1 RBD of the double mutant variant with hACE2. The findings possibly suggest the easier susceptible (lower binding energy score) and mutable traits (worse structural the interaction stabilities) of the double mutant variant. We also determined that the double mutations could weaken the effects of mAbs designed for WT by reducing the neutralizing activity, which may be the cause of the easily escapable nature of the double mutant variant. Etesevimab is found to be more effective in preventing the variant containing E484Q and L452R than casirivimab and bamlanivimab.

## Figures and Tables

**Figure 1 ijms-24-11564-f001:**
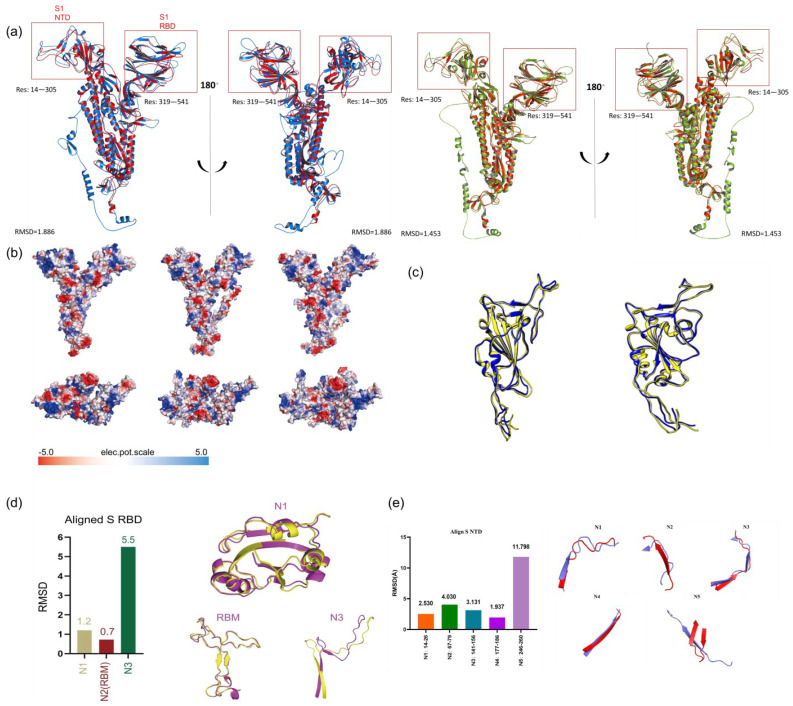
(**a**) AlphaFold2 alignment of predicted structures of S proteins of the double mutant variant and experimental WT (PDB: 7DF4). The predicted Rank 1, Rank 0, and experimental structures are colored in blue, green, and red, respectively. (**b**) The electrostatic surfaces of S protein, S1 RBD, and S1 NTD, with red and blue colors representing negative and positive charges, respectively. The WT, predicted Rank 1, and Rank 0 are presented from left to right. (**c**) AlphaFold2 alignment of predicted structures of S1 RBDs of the double mutant variant and WT (PDB: 6M0J). The predicted structures and experimental structures are colored in yellow and blue. The predicted Rank 0 and Rank1 are on the left and right. (**d**) Structural comparison of S1 RBDs of the double mutant variant and WT. The figure on the left shows the RMSD values of N1, RBM, and N3, while the structural comparison is shown on the right. (**e**) Comparison of five loops of S1 NTDs of the double mutant variant and WT. The RMSD values are shown on the left, while the structural comparisons are shown on the right.

**Figure 2 ijms-24-11564-f002:**
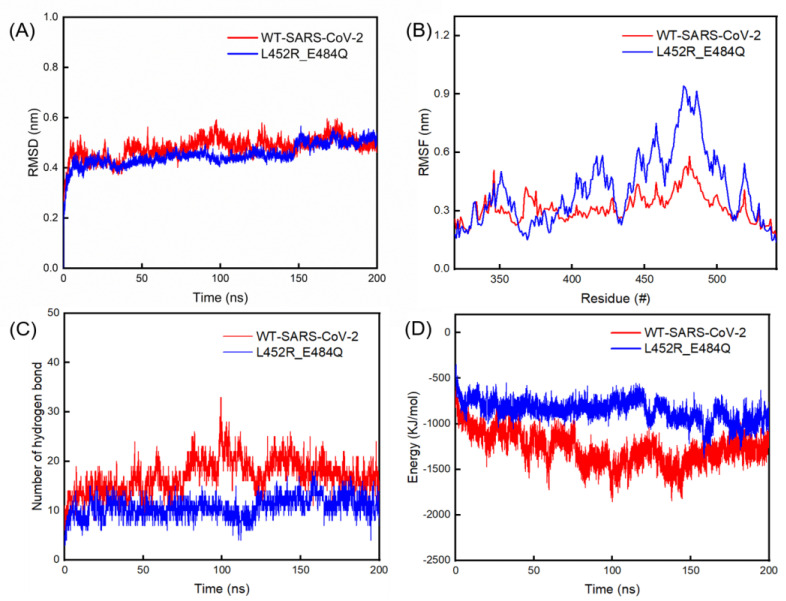
(**A**) Evolutions of the RMSD values of double mutant variant-hACE2 and WT-hACE2 complexes. (**B**) Evolutions of the RMSF values of WT and double mutant variant. (**C**) Number of hydrogen bonds and (**D**) interaction energy between the S proteins of WT and double mutant variant with hACE2.

**Figure 3 ijms-24-11564-f003:**
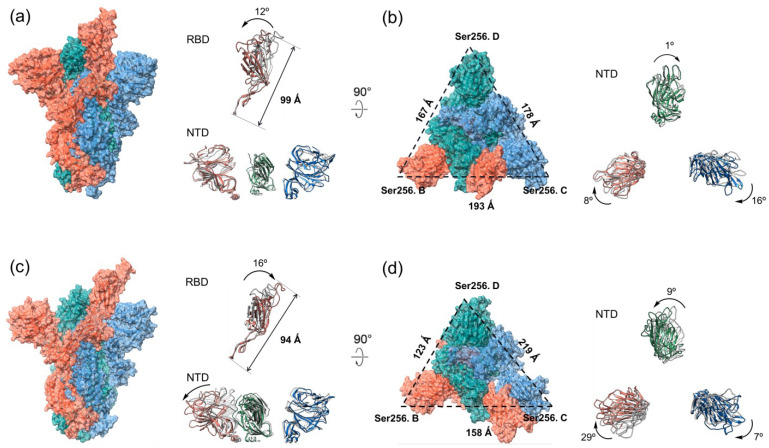
The architectures of the S proteins of WT and double mutant variant after the 200 ns MD simulations. The S1 RBD and S1 NTD structures of S proteins after 200 ns MD simulations are colored, while their structures before the simulation are in dark gray. The B chain, C chain, and D chain of S proteins are colored in red, blue, and green, respectively. (**a**) The side views of the S protein, S1 RBD, and S1 NTD of WT. (**b**) The top views of the S protein and S1 NTD of WT. (**c**) The side views of the S protein, S1 RBD, and S1 NTD of double mutant variant. (**d**) The top views of the S protein and S1 NTD of double mutant variant.

**Figure 4 ijms-24-11564-f004:**
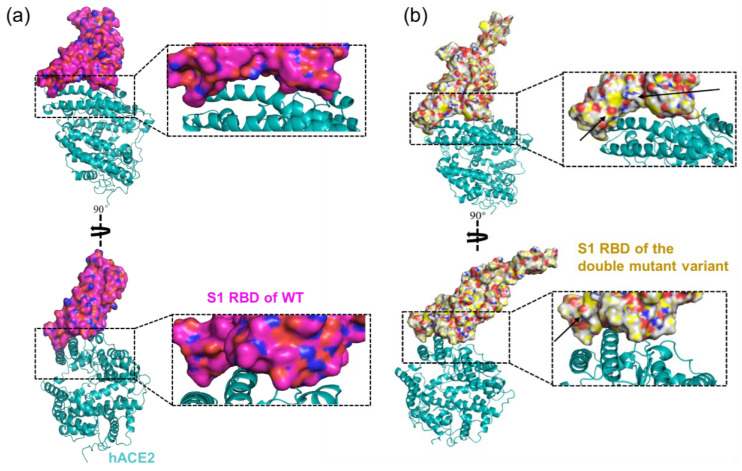
The interactions of (**a**) S1 RBD of WT and (**b**) S1 RBD of the double mutant variant with hACE2. The complex of S1 RBD of WT with hACE2 is downloaded from PDB (6M0J). The S1 RBD of the double mutant variant comes from AlphaFold2-predicted Rank 1 structure.

**Figure 5 ijms-24-11564-f005:**
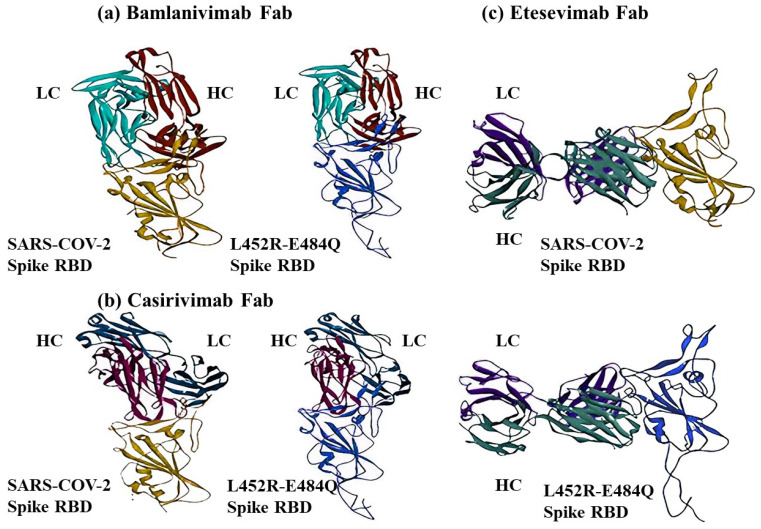
The interactions between S1 RBDs and mAbs. The S1 RBD of WT was downloaded and extracted from PDB (6M0J). The S1 RBD of double mutant variant comes from AlphaFold2-predicted Rank 1 structure. (**a**) HDOCK analyses of Bamlanivimab Fab against S1 RBDs of WT and double mutant variant. (**b**) HDOCK analyses of Casirivimab Fab against S1 RBDs of WT and double mutant variant. (**c**) HDOCK analyses of Etesevimab Fab against S1 RBDs of WT and double mutant variant. The S1 RBDs of WT and double mutant variant are colored dark yellow and light blue. The dark red and cyan represent the heavy chain (HC) and light chain (LC) of Fab. The HC and LC of Casirivimab Fab are colored in dark blue and fuchsia. The HC and LC of etesevimab Fab are colored dark green and purple.

**Table 1 ijms-24-11564-t001:** The docking scores of WT-hACE2 and double mutant variant-hACE2 complexes using HADDOCK and HDOCK.

	WT-hACE2	Double Mutant Variant-hACE2
**HADDOCK score**	−116.1 ± 1.6	−123.7 ± 4.0
**RMSD from the overall lowest energy**	6.5 ± 0.2	0.8 ± 0.5
**Van der Waals energy**	−56.4 ± 3.0	−63.6 ± 4.4
**Electrostatic energy**	−221.5 ± 11.0	−248.1 ± 29.2
**Buried surface area (Å^2^)**	1710.8 ± 41.9	1834.8 ± 27.4
**Z-score**	−1.3	−2.1
**HDOCK docking score**	−310.19	−347.29
**Ligand RMSD (Å)**	3.27	1.94

**Table 2 ijms-24-11564-t002:** The HADDOCK and HDOCK docking scores of WT and double mutant variant against three monoclonal antibodies.

Antibodies	S Protein	Docking Scores	
		HDOCK	HADDOCK
Bamlanvimab	WT	−125.996	−275.49
	Double mutant variant	−107.493	−181.94
Casirivimab	WT	−121.918	−407.74
	Double mutant variant	−94.374	−94.374
Etesevimab	WT	−123.423	−263.07
	Double mutant variant	−108.941	−184.8

## Data Availability

Not applicable.

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
