# Peer review of "Impact of E484Q and L452R Mutations on Structure and Binding Behavior of SARS-CoV-2 B.1.617.1 Using Deep Learning AlphaFold2, Molecular Docking and Dynamics Simulation"

_ijms, 2023, doi:10.3390/ijms241411564_

Round 1
Reviewer 1 Report
The Authors reported the structural investigation of SARS-CoV-2 mutated spike protein, interacting with hACE2. The idea is compelling, bu the discussion of the reults is quite rough
1) What do you mean with "5 rings". In the caption of Figure 1e are indicated as loops.
2) How did you calculate the interaction energy between hACE2-wt and hACE2-mutants complexes?
3) Pag. 4, line 158. What is the reference that you used for RMSD calculations?
4) The reference of CASP14 should be inserted, as descrbing the used parameters
5) Pag 10, line 333, correct "Glu484 to Gln484".
6) Molecular docking studies with mABs should be integrated with molecular dynamics simulations. This in light of opposite results obtained by two methodology by two mutants and WT structures.
7) The comparision of atom interactions by mutated residues and hACE2 couterparts should be described in details
8) Due to the resolution of deposited structure 7DF4 and the uncertainity associated to the Alhafold built model, a longer equilibration time is needed before molecular dynamics simulation. Maybe, these could justify the opposite reults between molecular docking and MD. Moreover, did you exceute MD replicas?
9) The Authors assesed the observation of lower H-bonds, but the manuscript lacks a detailed explanation.
10) How is the RMSD of docked complex respect to the experimental one?
11) The structural chenges described in the section 2.3 are interesting, but a detailed discussion on their origin should be reported.
12) The input structures of MD are the docked complexes?
Reviewer 2 Report
The manuscript of Yanqi Jiao et al. reports a study on the impact of two interesting mutations of SARS-CoV2. From a general point of view, the work is technically well done and the conclusions are reasonable both in the light of what is known about the behavior of the protein and on the basis of the computational results reported by the authors.
The only comment I would like to make is that, although the simulation is long enough to observe the dynamic behavior of the protein, currently (at least in the journals dedicated to computational biochemistry) there is a tendency to carry out some replicates of the same simulation (generally three to five). Since it seems to me that the authors are discussing only one simulation, if they could make even a minimal number of replicates, the statistical consistency would be much greater. Alternatively, it should be discussed and justified why the authors believe that what was obtained with the single simulation is reliable.
Minor points:
Page 2, lines 50-60: it would be better to use a single convention to indicate the mutations (eg T95I).
Table 1: even if in line with what has been reported, is van der Waals energy really positive?
Round 2
Reviewer 1 Report
The AUthors answered to the issues raised by this Reviewers. The manuscript deserves the acceptance.